# Searching for Antimicrobial-Producing Bacteria from Soils through an Educational Project and Their Evaluation as Potential Biocontrol Agents

**DOI:** 10.3390/antibiotics13010029

**Published:** 2023-12-28

**Authors:** Mario Sergio Pino-Hurtado, Rosa Fernández-Fernández, Carmen Torres, Beatriz Robredo

**Affiliations:** 1Area of Biochemistry and Molecular Biology, OneHealth-UR Research Group, University of La Rioja, 26006 Logroño, Spain; mario-sergio.pino@unirioja.es (M.S.P.-H.); rosa.fernandez@unirioja.es (R.F.-F.); carmen.torres@unirioja.es (C.T.); 2Area of Didactic of Experimental Sciences, OneHealth-UR Research Group, University of La Rioja, 26006 Logroño, Spain

**Keywords:** antimicrobial-producing bacteria, bacteriocin, biocontrol agents, fungi, MicroMundo project, soil

## Abstract

Antimicrobial resistance (AMR) is a serious threat to public health due to the lack of effective drugs to combat infectious diseases, which generates the need to search for new antimicrobial substances. In this study, the potential of soil as a source of antimicrobial-producing bacteria (APB) was investigated and the importance of the connection between education and science was emphasized, using service-learning methodologies. Sixty-one soil samples were collected, and 1220 bacterial isolates were recovered. Eighteen of these isolates showed antimicrobial activity against at least 1 of the 12 indicator bacteria tested (including multidrug-resistant and relevant pathogens). The 18 APB were identified by MALDI-TOF and 6 different genera (*Bacillus*, *Brevibacillus*, *Lysinobacillus*, *Peribacillus*, *Streptomyces*, and *Advenella*) and 10 species were identified. The 18 APB were tested for antifungal activity against four phytopathogenic fungi (*Botritis cynerea*, *Lecanicillium fungicola*, *Trichoderma harzianum*, and *Cladobotryum mycophilum*). Moreover, the antibiotic susceptibility of APB was tested using the disk-diffusion method as well as their β-hemolytic activity (important safety criteria for potential future applications). A total of 10 of the 18 APB were able to inhibit at least 50% of indicator bacteria tested, including methicillin-resistant *Staphylococcus aureus* (MRSA), among others. A total of 4 of the 18 APB (3 *Bacillus pumilus* and 1 *Bacillus altitudinis*) showed inhibitory activity against two of the four fungal pathogens tested (*B. cinerea* and *L. fungicola*), as well as against 5–7 of the 12 bacterial pathogen indicators; these 4 isolates showed susceptibility to the antibiotics tested and lacked β-hemolytic activity and were considered promising APB for use as potential biocontrol agents. In addition, one *Brevibacillus laterosporus* strain had activity against 83% of indicator bacteria tested including *Escherichia coli*, MRSA and other methicillin-resistant staphylococci, as well as vancomycin-resistant enterococci (but not against fungi). These results show that soil is a source of APB with relevant antibacterial and antifungal activities, and also emphasize the importance of education and science to raise public awareness of the AMR problem and the strategies to control it.

## 1. Introduction

Antimicrobial resistance (AMR) represents an increasing challenge for global public health due to the lack of effective drugs to combat infectious diseases, leading to the need to look for new antimicrobial substances [1]. The constant movement of microorganisms between plants, animals, and humans is key for maintaining the good health of all organisms within an ecosystem [2,3]. 

Soils are very important for global health and represent a huge reservoir for the immense diversity of microorganisms on our planet [4,5], and serve as a site for the exchange of substances between these microbial communities and plants [6,7]. The organisms most commonly found in soil are bacteria, which play an important role in these processes of exchange and transformation of substances and materials (decomposition of organic matter, transformation of soil nutrients, and regulation of soil fertility) [8,9,10]. Bacteriocins are bioactive peptides with antimicrobial properties that are ribosomally synthesized by a wide range of bacteria as a defense mechanism against other microorganisms with which they coexist in their ecosystem [11,12,13]. These peptides allow bacteria to increase their chances of adaptation in a hostile environment and, if the mechanisms involved can be understood, they are an excellent alternative for dealing with AMR [13,14,15].

For several years, the massive use of chemical pesticides has been causing harmful effects on the environment, especially on soils [16]. The search for ecological methods that are both environmentally friendly and economically feasible is currently the goal of many researchers [17]. Soil bacteria, which are constantly competing for limited resources, represent a very good starting point for the search for new molecules with antimicrobial activity and biocontrol potential [7,18]. Bacteria of the genera *Bacillus* and *Brevibacillus* are widespread in nature, mainly in soil, and have been found to produce many peptides with antibacterial and antifungal activities. Their use as biocontrol agents could control plant diseases or pests caused by bacteria, fungi, and viruses, as well as pathogenic yeasts and protozoa [19,20].

Moreover, the search for new antimicrobial substances is extended to society through educational citizen science projects, such as MicroMundo. The MicroMundo service-learning educational project teaches knowledge of soil microbiota through the search of antimicrobial substances while trying to raise awareness about the AMR problem [21]. MicroMundo is integrated into a global citizen science project in AMR called “Tiny Earth” [22], originally implemented in 2012, in the USA with “Small World Initiative” designation [23]. We highlight the benefits of global collaboration against AMR and advocate for participation in the OneHealth initiative to address and mitigate the challenges posed by this silent pandemic [24,25].

We have previous experience in the development of the MicroMundo project in the region of La Rioja (Spain) in 2019 [26] and 2020–2022 [27]. After the success achieved in those years in terms of citizen participation and awareness of the AMR problem, we continued the project in 2023, looking not only for antimicrobial-producing bacteria (APB), but also for those of interest as potential biocontrol agents. This study demonstrates the relevant link between science and education and the benefits of implementing service-learning methodologies with new students (secondary school, Masters in Education, and PhD students) in different educational institutions. Therefore, the present work aimed to search for APB (putative producers) from soils at new sampling points, and to determine their potential as antibacterial and antifungal biocontrol agents. 

## 2. Results

### 2.1. Evaluation of the Antibacterial Activity in the First Screening

A total of 61 soil samples were analyzed during the MicroMundo project, and 1220 bacterial isolates (20 isolates/sample) were obtained and subjected to a first screening for antimicrobial activity. Among them, 52 isolates (4.2%) showed potential inhibitory capacity in the initial school-level screening against two indicator bacteria (*E. coli* and *S. epidermidis*). The identification by MALDI-TOF of 51 of the 52 putative APB (1 isolate could not be identified) revealed that 42 were Gram-positive bacteria (22 species and 11 genera) and the other 9 isolates were Gram-negative bacteria (8 species and 4 genera) (Figure 1A). Overall, *Bacillus* was the most abundant genus of potential APB (34.6%). The following microbial diversity was detected: *Bacillus* (18 isolates), *Peribacillus* (7), *Pseudomonas* (6), *Lysinibacillus* (5), *Paenarthrobacter* (3), *Paenibacillus* (2), *Advenella* (1), *Brevibacillus* (1), *Escherichia* (1), *Exiguobacterium* (1), *Psychrobacillus* (1), *Scandinavium* (1), *Staphylococcus* (1), and *Streptomyces* (1) (Figure 1A).

### 2.2. Verification of Antibacterial Activity in the Second Screening

Subsequently, these 52 isolates were subjected to a second screening in the laboratory of the University of La Rioja. Thus, the antimicrobial activity was evaluated against 12 indicator bacteria using the Spot-on-Lawn method [27] and after rigorous repetitions, 18 out of the 52 bacteria isolated were finally selected because of their clear antimicrobial activity against at least one indicator bacteria and were considered as APB (Figure 1B and Table 1).

The APB were considered highly effective producers if they showed activity against more than 50% of the indicators tested, as was shown for six of the isolates: *B. laterosporus* X9433 (83%); *B. pumilus* X9430, X4969, and X9475 (58%); and *Bacillus thuringiensis* X4968 and X4970 (58%) (Table 1). All of these isolates inhibited methicillin-resistant *Staphylococcus aureus* (MRSA) and methicillin-resistant *Staphylococcus pseudintermedius* (MRSP) indicator isolates, and five of them also inhibited *Escherichia coli*. Four additional APB inhibited 50% of indicator bacterial isolates.

The most susceptible indicator bacteria to the action of the APB were as follows: *Staphylococcus delphini* (78% inhibition), MRSP (78%), *Micrococcus luteus* (78%), methicillin-susceptible *S. pseudintermedius* MSSP (72%), MRSA (61%), and methicillin-resistant *Staphylococcus epidermidis* (MRSE) (61%) (Table 1). 

Interestingly, five APB showed antimicrobial activity against the Gram-negative indicator *E. coli* (*B. laterosporus* X9433; *B. pumilus* X9430, X9469, and X9475; and *Bacillus subtilis* X9429). Moreover, *Brevibacillus laterosporus* X9433 showed antimicrobial activity against 10 out of the 12 indicator bacteria (83%) and was the only isolate able to inhibit *Listeria monocytogenes*, a very important food-borne pathogen, as well as to inhibit vancomycin-resistant *Enterococcus* isolates (Table 1).

### 2.3. Evaluation of the Antifungal Activity

To evaluate the potential of the 18 selected APB as biocontrol agents, four fungal pathogens were considered in this study. Figure 2 shows the results of the analysis for the determination of the antifungal activity of the 18 APB against the four fungi seeded in Czapek–Dox agar plates (Condalab, Madrid, Spain). 

In the plates inoculated with *B. cinerea* (Figure 2A), it was observed that isolates *B. pumilus* X9427, X9430, and X9475 and *B. altitudinis* X9472 showed strong antifungal activity against this pathogen. Furthermore, in the plates inoculated with *L. fungicola* (Figure 2B), the isolates *B. pumilus* X9426, X9428, X9430, X9431, X9469, X9474, and X9475, *B. altitudinis* X9472, and *L. fusiformis* X9474 also showed clear antifungal activity. For plates inoculated with the pathogens *T. harzianum* and *C. mycophilum*, there was no antifungal activity by any of the 18 APB (Figure 2C,D).

### 2.4. Safety Assessment of the Antimicrobial-Producing Isolates

#### 2.4.1. Antibiotic Susceptibility Testing of the APB

Antibiotic susceptibility testing was performed for the 15 APB of the genera *Bacillus Brevibacillus, Lysinibacillus*, and *Peribacillus*, using the breakpoints of *Bacillus* (EUCAST 2023). The remaining three APB isolates belong to the genera *Streptomyces* and *Advenella* (including also isolate X4973, which was not identified by MALDI-TOF), for which breakpoints could not be found in the guidelines and therefore, their antibiotic susceptibility was not analyzed. Table 2 shows the inhibition halos (mm) around the disc of the tested APB. As it is shown, all these bacteria showed susceptibility to all the antibiotics tested, except for *B. laterosporus* X9433 to ERY_15_ and *B. pumilis* X9430 to CLI_2_.

#### 2.4.2. β-Hemolytic Activity of the Antimicrobial-Producing Isolates

A total of 8 APB (3 *B. pumilus*, 1 *B. laterosporus*, 1 *S. prasinus*, 1 *P. muralis*, 1 *A. kashmirensis* and 1 *B. altitudinis*) lacked β-hemolytic activity, while the remaining 10 antimicrobial-producing isolates showed either weak (n = 5) or strong (n = 5) β-hemolytic activity (Table 3).

## 3. Material and Methods

### 3.1. Study Area and Sampling

MicroMundo was implemented in the region of La Rioja (Spain) in 2023 through two phases of practical training involving the University of La Rioja and five secondary education institutions. The first phase of the project was carried out at the University itself and was conducted by a qualified professor who trained secondary education Masters’ students (n = 22), doctoral students (n = 4), and other biology teachers (n = 9). During the training sessions, the methodology and logistics of the planned sampling points in the different geographical locations of the region were discussed. The second phase of the project was carried out in secondary schools and involved postgraduate students, secondary school teachers, and 177 secondary school students. A total of 61 groups were formed, and each of them analyzed one soil sample.

### 3.2. First School-Level Screening during the MicroMundo Project

The 61 soil samples were first diluted and plated onto tryptic soy agar plates (Condalab, Madrid, Spain) for selection of bacterial colonies (20 isolates/sample) according to the methodology proposed by Robredo et al. [26]. The antimicrobial activity of the 20 isolates obtained from each soil sample was tested, using Gram-positive *Staphylococcus epidermidis* C2663 and Gram-negative *Escherichia coli* C408 as indicator bacteria. The indicator bacteria were resuspended in saline and plated onto tryptic soy agar plates in a lawn pattern. Microbial isolates to be tested for the production of antimicrobial activity were then transferred using a sterile toothpick. After 24 h of incubation, the plates were evaluated by the students and the isolates with potential inhibitory activity (presence of inhibition halos) were selected and sent to the university for further verification and characterization.

### 3.3. Bacterial Identification

Matrix-assisted laser desorption/ionization–time of flight mass spectrometry (MALDI-TOF) was used to identify the isolates with potential antimicrobial activity in the first screening (n = 52). The recommended standard protein extraction protocol for the commercial device from Bruker Daltonics, Germany, was followed (MALDI-TOF Biotyper^®^, Bruker, Billerica, MA, USA).

### 3.4. Evaluation of Antibacterial Activity 

#### Second Screening of Antibacterial Activity Using the Spot-on-Lawn Method

The selected isolates that showed potential antibacterial activity in the first school-level screening (n = 52) were subjected to further analysis and characterization at the university. To achieve this, the isolates were tested against 12 indicator bacteria including multidrug-resistant and relevant pathogens.

The bacteria were cultured on brain heart infusion agar plates (Condalab, Madrid, Spain) and incubated at 37 °C for 24 h to obtain a pure culture. To prepare the test plates, a suspension of the indicator strain was prepared in a 3 mL sterile saline solution to give a turbidity of 0.5 MacFarland. This suspension was spread onto solid tryptic soy agar plates (supplemented with 0.3% yeast extract) using a sterile swab (Condalab, Madrid, Spain).

A fresh solid culture of each isolate being tested for antimicrobial activity was spotted onto the agar plates seeded with the indicator strains using a sterile toothpick. The plates were incubated at 37 °C for 24 h to allow for the evaluation of inhibition zones. If a bacterial isolate showed a clear and distinct inhibition zone against at least 1 of the 12 indicator strains, it was classified as an antimicrobial-producing bacteria (APB).

### 3.5. Evaluation of Antifungal Activity 

The antifungal activity of the APB obtained in the second screening (n = 18) was evaluated against four fungi considered to be relevant phytopathogens: (a) *Botrytis cinerea*, which is involved in gray rot in grapevines; (b) *Lecanicillium fungicola* and (c) *Trichoderma harzianum*, which are two pathogenic fungi of the white mushroom (*Agaricus bisporus*); and (d) *Cladobotryum mycophilum*, which is the causal agent of cobwebs in mushroom crops. These fungi are responsible for significant losses in the agricultural sector. 

#### 3.5.1. Preparation of the Inoculum (Conidial Suspension)

A culture of each fungus was prepared on potato dextrose agar plates (Condalab, Madrid, Spain) for 7 days at 25 °C. Each plate was washed with 10 mL of sterile distilled water, and then scraped with a seeding loop to release the conidia. This suspension was filtered with a sterile Miracloth Merk polypropylene mesh filter (20–25 µm pore diameter) to remove any mycelial fragments. The conidial concentration of the suspension was measured with a hemocytometer and adjusted with sterile distilled water to the required concentration (7.5 × 10^3^ spores/mL).

#### 3.5.2. Seeding of the Conidial Suspension 

A 25 µL volume of the conidial suspension of each of the four fungi was seeded in triplicate on Czapek–Dox agar plates (Condalab, Madrid, Spain) using a Drigalsky loop. This medium was chosen because it was the most suitable for the growth of both fungi and APB.

#### 3.5.3. Inoculation of Antimicrobial-Producing Isolates

Three plates were inoculated with each fungus, and later, 18 sterile discs were placed in two of these plates seeded with 10 µL of an inoculum of the 18 APB (concentration of 2 McFarland). The third plate was used as a fungal growth control (C+), and no bacterial strain was inoculated. The results were measured after 7 days of incubation at 25 °C. 

### 3.6. Safety Assessment of Antimicrobial-Producing Isolates

#### 3.6.1. Antibiotic Susceptibility Testing 

Antibiotic susceptibility testing using the disk diffusion method was performed on the APB corresponding to genera for which do exist breakpoints in international committees. In this sense, the European Committee on Antimicrobial Susceptibility Testing guidelines (EUCAST, 2023) were followed for interpreting the results for *Bacillus* spp. and related genera (*Brevibacillus, Lysinibacillus*, and *Penibacillus*); the APB of the remaining genera were not tested for antibiotic susceptibility testing due to lack of standard breakpoints. For this assay, a standard inoculum of 0.5 McFarland of the bacteria was applied to the surface of Muller–Hinton plates (Condalab, Madrid, Spain). Disks of the antibiotics (OXOID) were placed on the agar surface. After an incubation period of 24 h at 37 °C, the diameter of the inhibition zone around the disc was measured. The antibiotics tested were as follows (abbreviation and dose in mg): imipenem (IMP_10_), erythromycin (ERY_15_), clindamycin (CLI_2_), ciprofloxacin (CIP_5_), and linezolid (LZD_10_).

#### 3.6.2. β-Hemolytic Activity Test

This test was used to evaluate the ability of the APB (n = 18) to exhibit β-hemolytic activity. During the test, the isolated bacteria were spotted with sterile toothpicks on blood agar plates (Condalab, Madrid, Spain) and were incubated under appropriate conditions. Bacterial colonies were observed for the presence or absence of complete hemolysis (beta-hemolysis). This assay was conducted with the objective of determining their suitability as potential bioprotective agents in agriculture.

## 4. Discussion

This research carried out extensive soil sampling thanks to citizen collaboration. A total of 1220 bacteria were isolated, of which, 18 APB were identified, representing 1.5% of the total recovered isolates. These results are similar to those carried out during the former study conducted by Fernández-Fernández et al. 2022 [27], in which only 1.2% of total tested isolates were selected as potential antimicrobial producers.

It is important to ensure the absence of acquired antibiotic resistance mechanisms or virulence factors in the strains that are candidates for potential biocontrol agents [28]. Therefore, in this work, antibiotic susceptibility tests were carried out on 15 of the 18 APB, using the breakpoints for *Bacillus* spp., and demonstrated that they were susceptible to the antibiotics tested with a few exceptions. In addition, eight of them lacked β-hemolytic activity, including isolates with high inhibitory activity against multidrug-resistant (MDR) bacteria (*B. laterosporus*) and isolates with activity against fungi (*B. pumilus* and *B. altitudinis*), which makes them promising candidates for potential future applications. The members of *Bacillus* and *Brevibacillus* genera obtained in this research are known for their ability to produce a variety of bioactive compounds with different activities such as antibacterial [29,30], antifungal [16,31,32,33,34,35,36,37,38], antiviral [39,40,41], and pest control properties [42,43].

*Bacillus* spp. has been widely used to control plant diseases such as wilt in tomato [36], tobacco [39], and banana [37], which are all caused by fungi, as well as bacterial wilt in tobacco [38]. Members of the genus *Bacillus* have been most commonly chosen to prepare bioformulations with a positive impact on soil health as well as plant growth and health [44,45,46]. Regarding *B. laterosporus*, Chen et al. 2017 [47] made the first report on the effective biocontrol of potato common scab (an economically important disease caused by *Streptomyces* spp.) using *B. laterosporus*. Li et al. 2021 [48] demonstrated that the application of the strain *B. laterosporus*, used for effective control of potato common scab, successfully modified the composition and function of soil bacterial community in the tuberosphere and rhizosphere.

*Bacillus subtilis* is currently considered one of the most promising microorganisms in sustainable agriculture [49] and it has been reported as a growth promoter and has activity against a wide variety of plant pathogens [50], although in our study, the only strain of *Bacillus subtilis* had β-hemolytic activity and was not a good candidate. Lahlali et al. 2013 [51] have reported the successful application of the biofungicide Serenade (*B. subtilis*), which was effective against infection of canola by *Plasmodiophora brassicae*. It has been demonstrated that *B. subtilis*, isolated from the rhizosphere of tomato plants, inhibited the growth of *Fusarium oxysporum* and *R. solanacearum*, the main phytopathogens that hinder this type of greenhouse crop [44]. 

However, we have detected three isolates of *B. pumilus* with potential as biocontrol agents, the same species successfully used by Dai et al. in 2021 [52] to inhibit the growth of *Sphaeropsis sapinea*, the pathogen responsible for pine shoot blight disease. 

In our study, *B. laterosporus* X9433 is particularly remarkable for its broad spectrum of action against 10 of the 12 indicator bacteria (83%) including MRSA, MRSE, MRSP, *L. monocytogenes*, *E. coli*, and vancomycin-resistant *Enterococcus* isolates, although it did not show antifungal activity. This strain will be studied by whole-genome sequencing in future assays to determine whether it possesses genes encoding brevibacillin (antimicrobial lipopeptide effective against multidrug-resistant strains) [32,53,54,55,56,57]. 

Authors such as Zhao et al. 2021 [55] succeeded in identifying, purifying, and characterizing a brevibacillin 2V from *B. laterosporus* with strong antimicrobial activity against *E. faecium, E. faecalis*, and MRSA, while Yang et al. 2018 [57] studied the antimicrobial mechanism of action of brevibacillin against sensitive *S. aureus* strains, although further studies are still needed to better understand its antimicrobial capacity. In addition, Le Han et al. 2022 [32] isolated strains of *B. halotolerans* and *B. laterosporus* from marine sediment samples that produced thermostable and pH-tolerant antifungal compounds, which inhibited the growth of relevant pathogens such as *Alternaria alternata*, *Candida albicans*, *Cladosporium* sp., *Trichophyton rubrum*, *Trichosporon pullulans*, and *F. oxysporum*. The data obtained in this study corroborate the antibacterial capacity of *B. laterosporus*, which has also been previously detected in soil samples by our research group [27].

In summary, the *B. pumilus* X9430, X9469, and X9475, *B. subtilis* X9429 *and B. laterosporus* X9433 isolates recovered in this study showed activity against the Gram-negative indicator *E. coli*, and four of them also inhibited MRSA, MRSE, and MRSP (except *B. subtilis* X9429). Moreover, the isolates *B. pumilus* X9430 and *B. altitudinis* X9472 expressed antifungal activity against *B. cinerea* and *B. pumilus* X9426, X9430, and X9431 showed significant antimicrobial activity against *L. fungicola*.

## 5. Conclusions

This study highlights the threat of AMR and the need to find new solutions from a “One Health” perspective. The antimicrobial activity of selected soil bacteria was evaluated to identify potential biocontrol agents. A total of 3 of the 18 APB were considered excellent candidates for further studies to fully determine their potential as biocontrol agents, as they met the eligibility criteria established in this study (lack of β-hemolytic activity and absence of resistance to the tested antibiotics): (a) *B. pumilus* X9430 and (b) *B. altitudinis* X4972 (which inhibited three methicillin-resistant *Staphylococcus* indicator bacteria and two fungal pathogens); (c) *B. laterosporus* X9433 (with strong antibacterial activity against MDR bacteria). These results underscore the potential of soil bacteria as part of the solution to AMR and the importance of community engagement in the fight against this global health challenge.

## Figures and Tables

**Figure 1 antibiotics-13-00029-f001:**
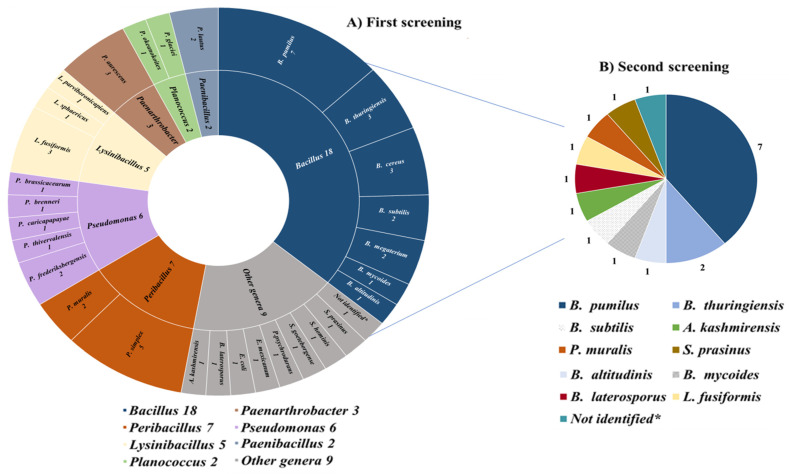
Genus and species identification of (**A**) the 51 potential APB obtained in the first screening; (**B**) the 18 APB verified in the second screening. * One of the isolates could not be identified by MALDI-TOF.

**Figure 2 antibiotics-13-00029-f002:**
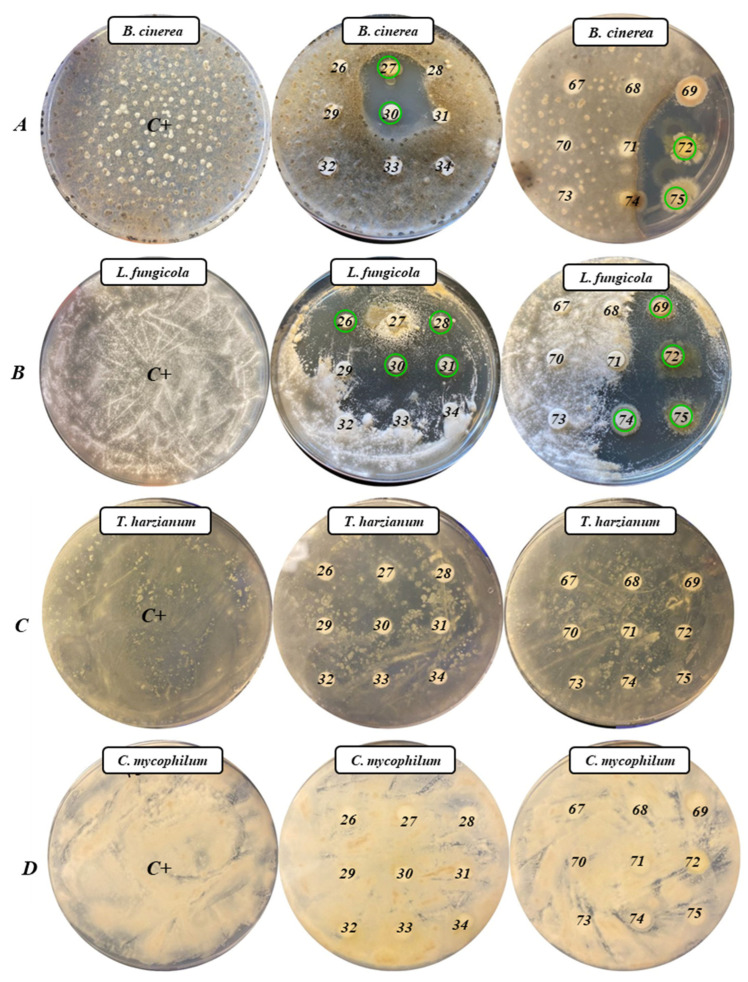
Antifungal activity of the 18 APB in Czapek–Dox Agar plates ((**A**): *B. cinerea*; (**B**): *L. fungicola*; (**C**): *T. harzianum*; and (**D**): *C. mycophilum*). Green circles mark the isolates with positive activity. Isolates tested were 26: X9426; 27: X9427; 28: X9428; 29: X9429; 30: X9430; 31: X9431; 32: X9432; 33: X9433; 34: X9434; 67: X9467; 68: X9468; 69: X9469; 70: X9470; 71: X9471; 72: X9472; 73: X9473; 74: X9474; 75: X9475.

**Table 1 antibiotics-13-00029-t001:** Antibacterial activity profile of the 18 APB against the 12 indicator bacteria tested (in green: positive; in red: negative).

Antibacterial Activity on the Indicator Bacteria														
					Indicator Bacteria									
Antimicrobial-Producing Isolates		*E. coli*	MRSE	MRSA	*M.* *sciuri*	*S.* *delphini*	MRSP	MSSP	*E. faecium* *van*R ^b^	*E.* *cecorum*	*E.* *faecalis*	*M.* *luteus*	*L.* *monocytogenes*	N (%) ^c^
Species	ID Number													
*Advenella kashmirensis*	X9471													6 (50)
*Bacillus altitudinis*	X9472													6 (50)
*Bacillus mycoides*	X9467													6 (50)
*Bacillus pumilus*	X9426													6 (50)
*Bacillus pumilus*	X9427													6 (50)
*Bacillus pumilus*	X9428													6 (50)
*Bacillus pumilus*	X9430													7 (58)
*Bacillus pumilus*	X9431													5 (42)
*Bacillus pumilus*	X9469													7 (58)
*Bacillus pumilus*	X9475													7 (58)
*Bacillus subtilis*	X9429													4 (33)
*Bacillus thuringiensis*	X9468													7 (58)
*Bacillus thuringiensis*	X9470													7 (58)
*Brevibacillus laterosporus*	X9433													10 (83)
*Lysinibacillus fusiformis*	X9474													1 (8)
*Peribacillus muralis*	X9434													1 (8)
*Streptomyces prasinus*	X9432													1 (8)
NI ^a^	X9473													2 (16)
Indicator Bacteria Inhibited (%) ^d^		5 (28)	11 (61)	13 (72)	7 (39)	14 (78)	14 (78)	13 (72)	1 (6)	1 (6)	14 (78)	1 (6)	1 (6)	

Abbreviations: MRSE (methicillin-resistant *Staphylococcus epidermidis*); MRSA (methicillin-resistant *Staphylococcus aureus)*; MRSP (methicillin-resistant *Staphylococcus pseudintermedius)*; MSSP (methicillin-susceptible *S. pseudintermedius)*; *E. coli*: *Escherichia coli*; *M. sciuri*: *Mammaliicoccus sciuri*; *S. delphini*: *Staphylococcus delphini*; *E. faecium*: *Enterococcus faecium*; *E. cecorum*: *Enterococcus cecorum*; *E. faecalis*: *Enterococcus faecalis*; *M. luteus*: *Micrococcus luteus*; *L. monocytogenes*: *Listeria monocytogenes.* ^a^ NI: not identified by MALDI-TOF. ^b^
*van*R: vancomycin-resistant indicator. ^c^ N (%): number and percentage of indicator bacteria inhibited by one APB (in green color). ^d^ Indicator bacteria inhibited (%): number and percentage of the APB inhibiting one indicator bacteria (in green color).

**Table 2 antibiotics-13-00029-t002:** Inhibition halos (mm) of the antibiotics tested in the 15 APB (EUCAST 2023 breakpoints for *Bacillus* spp.).

Species ^a^	ID Number	Antibiotic Tested				
		IPM_10_	ERY_15_	CLI_2_	CIP_5_	LZD_10_
*Bacillus altitudinis*	X9472	40	29	24	26	27
*Bacillus mycoides*	X9467	36	32	24	30	34
*Bacillus pumilus*	X9426	36	30	22	30	28
*Bacillus pumilus*	X9427	40	28	28	26	28
*Bacillus pumilus*	X9428	40	28	28	26	30
*Bacillus pumilus*	X9430	38	28	14	28	16
*Bacillus pumilus*	X9431	36	30	20	30	28
*Bacillus pumilus*	X9469	17	22	18	22	26
*Bacillus pumilus*	X9475	30	20	18	24	28
*Bacillus subtilis*	X9429	38	40	38	30	38
*Bacillus thuringiensis*	X9468	30	34	28	25	25
*Bacillus thuringiensis*	X9470	28	28	22	32	30
*Brevibacillus laterosporus*	X9433	40	16	30	20	28
*Lysinibacillus fusiformis*	X9474	30	28	28	24	29
*Peribacillus muralis*	X9434	38	35	36	32	34

^a^ *Streptomyces prasinus* X9432, *Advenella kashmirensis* X9471, and the unidentified isolate X9473 were not included because there are no EUCAST 2023 breakpoints.

**Table 3 antibiotics-13-00029-t003:** β-Hemolytic activity of the 18 antimicrobial-producing isolates.

Species	ID Number	β-Hemolytic Activity
*Advenella kashmirensis*	X9471	-
*Bacillus altitudinis*	X9472	-
*Bacillus mycoides*	X9467	++
*Bacillus pumilus*	X9426	-
*Bacillus pumilus*	X9427	+
*Bacillus pumilus*	X9428	+
*Bacillus pumilus*	X9430	-
*Bacillus pumilus*	X9431	-
*Bacillus pumilus*	X9469	++
*Bacillus pumilus*	X9475	+
*Bacillus subtilis*	X9429	+
*Bacillus thuringiensis*	X9468	++
*Bacillus thuringiensis*	X9470	+
*Brevibacillus laterosporus*	X9433	-
*Lysinibacillus fusiformis*	X9474	++
*Peribacillus muralis*	X9434	-
*Streptomyces prasinus*	X9432	-
NI*	X9473	++

(-): no inhibition halo (no β-hemolytic activity). (+): inhibition halo <15 mm (weak β-hemolytic activity). (++): inhibition halo > 15 mm (strong β-hemolytic activity). NI*: not identified by MALDI-TOF.

## Data Availability

The data presented in this study are available in the article.

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
