# Peer review of "Searching for Antimicrobial-Producing Bacteria from Soils through an Educational Project and Their Evaluation as Potential Biocontrol Agents"

_antibiotics, 2023, doi:10.3390/antibiotics13010029_

Round 1

Reviewer 1 Report

Comments and Suggestions for Authors

The authors of the study took up the important topic of searching for microorganisms that can be used in the fight against crop-destroying pathogens. The project involved participants (students, high school teachers, students and lecturers) who were able to collaborate in the McroMundo project funded in part by the Spanish government.

The project found 18 microorganisms that could potentially be used in agriculture to protect crops. The microorganisms were isolated from 61 soil samples. The research described here combines the microbiological aspect, the social aspect - by involving a large group of participants,  and the utilitarian aspect, because , after further research, it could lead to finding natural allies in crop protection. Presented investigations raise public awareness of the prevalence of antibiotic resistance.

A deficiency in the work is the numbering of chapters, which must be corrected before publication.

Author Response

Dear Reviewer 

Thanks to your revision, the manuscript has improved considerably.

Answer: We have changed the numbering and we have reordered the chapters. In methodology, 2.5. is now 2.6. and vice versa. In results, 3.3. is now 3.4. and vice versa.

Reviewer 2 Report

Comments and Suggestions for Authors

In this article, the authors used a very interesting approach to isolate potential antimicrobial producing bacteria using citizen-science. It is a well written article and minor revision is suggested.

Line 22: the prhase "promising due to their promising ability to inhibit two relevant pathogenic fungi", should be: "promising due to their in vitro ability to inhibit two relevant phytopathogenic fungi" in order to mislead the readers into the idea that these bacteria have the potential to inhibit human/animal pathogens also. And, why the inhibition of these pathogens were not tested?

Line 79: antifungal.

Line 194: 1,220

Lines 251-253. The prhase is missing something with the use of "moreover". Please, revise this.

The name of some microorganisms is not in italic (Line 291).

Line 295: [38], B. subtilis

Since the authors evaluated the antifungal activity of all 18 bacteria, the safety tests should be the last to be presented and discussed. Why perform the antifungal activity if some od the bacteria may not be suitable for their antimicrobial resistance and hemolytic actitivity? The AMR test is not clear, and the Table S1 should be included in the main text, although I did not have access to it. 

Line 398-399: "also be taken into consideration", it would sound better "also be further investigated regarding their potential"

Author Response

Dear Reviewer

Thanks to your revision, the manuscript has improved considerably. We respond your suggestions bellow.

Line 22: the phrase "promising due to their promising ability to inhibit two relevant pathogenic fungi", should be: "promising due to their in vitro ability to inhibit two relevant phytopathogenic fungi" in order to mislead the readers into the idea that these bacteria have the potential to inhibit human/animal pathogens also. And, why the inhibition of these pathogens were not tested?´

Answer: We have changed the sentence as indicated by reviewer.

Answer: The inhibition of these pathogens was tested, as it is shown in Table1.

Line 79: antifungal. Answer: It has been changed.

Line 194: 1,220. Answer:  It has been changed.

Lines 251-253. The phrase is missing something with the use of "moreover". Please, revise this. Answer: It has been changed.

The name of some microorganisms is not in italic (Line 291).

Answer: We have corrected and now all names of microorganisms are in italics.

Line 295: [38], B. subtilis. Answer: It has been changed.

Since the authors evaluated the antifungal activity of all 18 bacteria, the safety tests should be the last to be presented and discussed.

Answer: We have changed the numbering and we have reordered the chapters. In methodology 2.5. is now 2.6. and vice versa. In results 3.3. is now 3.4. and vice versa.

Why perform the antifungal activity if some of the bacteria may not be suitable for their antimicrobial resistance and hemolytic activity?

Answer: Indeed, it would not be necessary, but it was done with the 18 antimicrobial-producing isolates.

The AMR test is not clear, and the Table S1 should be included in the main text, although I did not have access to it.

Answer: We have included this table in the main text as suggested (new Table 2).

Line 398-399: "also be taken into consideration", it would sound better "also be further investigated regarding their potential"

Answer: It has been changed.

Reviewer 3 Report

Comments and Suggestions for Authors
  1. The paper titled “Searching for antimicrobial-producing bacteria from soils through an educational project: their potential as antifungal bi-ocontrol agents investigated the potential of soil as a source of antimicrobial-producing bacteria and importance of connection between education and science was explored, using service-learning methodologies. The manuscript has potential but need extensive changes before consideration
  2. Title is not describing the study as looks review article title so revise it as per specific objective of the study
  3. Abstract is written poor as lack the first sentence “ Antimicrobial resistance (AMR) is a serious threat to public health due to the lack of effective drugs to combat infectious diseases, which generates the need to search for new alternatives” is open ended  and what does here mean alternatives?. Also, the last sentence should be more clear in terms of the study findings focus like which is more important such as soil bacteria or connection between education and science. Also, methodology is lacking with study findings stakeholders for which this paper is useful at the end of abstract  
  4. Introduction looks fine
  5. In methodology, elaborate the procedure in detail for  Bacterial identification by MALDI-TOF
  6. Do not write any abbreviations in heading like “MALDI-TOF
  7. Add reference for the “Inoculation of antimicrobial-producing isolates”
  8. In methodology, the heading “Ethical issues” is inappropriate so need to replace
  9. In methodology, statistical analysis is missing?
  10. Results are written good however, add latest references for study support
  11. In conclusion major focus should be on findings with practical application for stakeholders and add some numerical results
Comments on the Quality of English Language
  1. In the manuscript, grammatical mistakes observed on few places so there is need to go through the paper for language and grammatical mistakes

Author Response

Dear Reviewer

Thanks to your revision, the manuscript has improved considerably. We respond your suggestions bellow.

Title is not describing the study as looks review article title so revise it as per specific objective of the study

Answer: We have changed the title: Searching for antimicrobial-producing bacteria from soils through an educational project and their evaluation as potential biocontrol agents.

Abstract is written poor as lack the first sentence “ Antimicrobial resistance (AMR) is a serious threat to public health due to the lack of effective drugs to combat infectious diseases, which generates the need to search for new alternatives” is open ended  and what does here mean alternatives?

Answer: We have changed the first sentence of the abstract: Antimicrobial resistance (AMR) is a serious threat to public health due to the lack of effective drugs to combat infectious diseases, which generates the need to search for new antimicrobial substances.

Also, the last sentence should be more clear in terms of the study findings focus like which is more important such as soil bacteria or connection between education and science. Also, methodology is lacking with study findings stakeholders for which this paper is useful at the end of abstract .

Answer: It has been modified: These results suggest that soil bacteria could be a potential solution to address AMR by the lack of effective antimicrobial drugs and, also emphasized the importance of education and science to raise public awareness of this problem.

Introduction looks fine

In methodology, elaborate the procedure in detail for  “ Bacterial identification by MALDI-TOF” Answer: The protocol used was the one cited in the text in lines 101-106.

Do not write any abbreviations in heading like “MALDI-TOF”.

Answer: MALDI-TOF has been removed from the heading.

Add reference for the “Inoculation of antimicrobial-producing isolates”.

Answer: We have used our own protocol therefore it is described in methodology and there is not a reference.

In methodology, the heading “Ethical issues” is inappropriate so need to replace

Answer: We have removed 2.7. because it is written at the end of the manuscript “Institutional Review Board Statement”

In methodology, statistical analysis is missing?

Answer: Your comment is interesting, but this study was not designed to perform a statistical analysis, but to obtain a collection of antimicrobial-producing isolates that could be effective against bacterial pathogens and fungi, which will be further evaluated in future studies.

Results are written good however, add latest references for study support.

Answer: These latest references are included.

In conclusion major focus should be on findings with practical application for stakeholders and add some numerical results.

Answer: We fully agree with this comment. It has been modified.

Comments on the Quality of English Language

In the manuscript, grammatical mistakes observed on few places so there is need to go through the paper for language and grammatical mistakes.

Answer: We have done a complete grammatical review of the manuscript.

Reviewer 4 Report

Comments and Suggestions for Authors

This manuscript is interesting. Nevertheless, it requires revisions to be publishable.

1. The educational project, although important, is not the matter of the article. I suggest only minor comments about it, focusing on the antimicrobial activity, which is the main subject.

2. Authors must reorganize the article to separate the two aspects covered by it: the antibacterial activity and the antifungal activity.

3. The sections regarding the antimicrobial activity of Bacillus and Brevibacillus must not be included in the Discussion section but in the Introduction section. Be as concise as possible in this aspect.

4. Center the Discussion comments on the results obtained. Organize the discussion part, emphasizing the results obtained and their importance.

5. Taking into account the form the article is redacted, it is not clear how the results could be applied to human pathogenic bacteria. In this case, the future isolation of bioactive compounds regarding this activity is worth to be discussed.

6. The table provided is not well structured for a clear understanding of the results presented. 

Comments on the Quality of English Language

The English language requires only minor revision.

Author Response

Dear Reviewer

Thanks to your revision, the manuscript has improved considerably. We respond your suggestions bellow.

  1. The educational project, although important, is not the matter of the article. I suggest only minor comments about it, focusing on the antimicrobial activity, which is the main subject.

Answer: We fully agree with this comment. We have modified the manuscript to highlight the antimicrobial activity part.

  1. Authors must reorganize the article to separate the two aspects covered by it: the antibacterial activity and the antifungal activity.

Answer: We have changed the numbering and we have reordered the chapters considering two sections: the antibacterial activity and the antifungal activity.

  1. The sections regarding the antimicrobial activity of Bacillus and Brevibacillus must not be included in the Discussion section but in the Introduction section. Be as concise as possible in this aspect.

Answer: We fully agree with this comment. It has been changed.

  1. Center the Discussion comments on the results obtained. Organize the discussion part, emphasizing the results obtained and their importance.

Answer: It has been changed, as indicated.

  1. Taking into account the form the article is redacted, it is not clear how the results could be applied to human pathogenic bacteria. In this case, the future isolation of bioactive compounds regarding this activity is worth to be discussed.

Answer: We fully agree with this comment. It has been changed. We have done a review of the manuscript. We have changed the numbering and reorder the chapters considering two sections: the antibacterial activity and the antifungal activity. The future assays for characterize bioactive compounds (brevibacillin) regarding this activity is briefly discussed in lines 319-321.

  1. The table provided is not well structured for a clear understanding of the results presented.

Answer: The table has been restructured for better understanding.

Reviewer 5 Report

Comments and Suggestions for Authors

In this research paper, authors isolated antimicrobial-producing bacteria from the samples and antifungal activity was analyzed. Further, they identified bacteria using MALDI-TOF and haemolytic activity was determined among bacteria. This is very interesting paper and is suitable for publication in this journal after minor revision.

I have some comments on this manuscript

Authors tested haemolytic activity of bacteria and the methodology and the result should be more clearer about it. I suggest using three groups, α-hemolytic (strong haemolytic activity), β-hemolytic (weak haemolytic group) and non-hemolytic group. If possible please classify the data accordingly.

In this research, authors used phtopathogens and only antibacterial activity, and antifungal activity were determined. But the biocontrol activity was originally not tested in vitro (for example seed germination assay in the laboratory), or in vivo (infection with plant pathogen, and treatment with bacteria or metabolites to improve the growth profile or to control disease caused by fungi). So I suggest modifying the title and pleased delete “Biocontrol” from the title and changing throughout the manuscript.

References are not according to the journal format, especially Journal abbreviation is not uniform. Please check and correct.

Author Response

Dear Reviewer

Thanks to your revision, the manuscript has improved considerably. We respond your suggestions bellow.

Authors tested haemolytic activity of bacteria and the methodology and the result should be more clearer about it. I suggest using three groups, α-hemolytic (strong haemolytic activity), β-hemolytic (weak haemolytic group) and non-hemolytic group. If possible please classify the data accordingly.

Answer. In this study β-hemolysis has been considered, the isolates with this activity have been indicated and they have been classified according to the diameter of the halo. For a better understanding of the results, the table 3 has been modified. (lines 265-273)

In this research, authors used phytopathogens and only antibacterial activity, and antifungal activity were determined. But the biocontrol activity was originally not tested in vitro (for example seed germination assay in the laboratory), or in vivo (infection with plant pathogen, and treatment with bacteria or metabolites to improve the growth profile or to control disease caused by fungi). So I suggest modifying the title and pleased delete “Biocontrol” from the title and changing throughout the manuscript.

Answer: Thanks for your comments. We understand your point, nevertheless, we think we could maintain the word "biocontrol" because we are talking about the "potential" that the isolates may have as biocontrol agents, not that they are already biocontrol agents. We have revised the manuscript to make clear this point to avoid confusion in this respect. Indeed, we will try to address this research in the future, as we mention now  in the new version of the manuscript (lines 344-346 have been modified).

References are not according to the journal format, especially Journal abbreviation is not uniform. Please check and correct.

Answer: We have changed the references according to the journal format.

Round 2

Reviewer 3 Report

Comments and Suggestions for Authors

All suggested changes are incorporated so paper may be considered for acceptance

Comments on the Quality of English Language

The language of the manuscript is acceptable 

Reviewer 4 Report

Comments and Suggestions for Authors

The authors have fulfilled the recommendations made.